# Probing spin dynamics of ultra-thin van der Waals magnets via photon-magnon coupling

Christoph W. Zollitsch [1] ✉, Safe Khan[1], Vu Thanh Trung Nam[2], Ivan A. Verzhbitskiy[2], Dimitrios Sagkovits[1,3], James O'Sullivan[1], Oscar W. Kennedy[1], Mara Strungaru[4], Elton J. G. Santos [4,5], John J. L. Morton [1,6], Goki Eda [2,7,8] & Hidekazu Kurebayashi [1,6,9]

Layered van der Waals (vdW) magnets can maintain a magnetic order even down to the single-layer regime and hold promise for integrated spintronic devices. While the magnetic ground state of vdW magnets was extensively studied, key parameters of spin dynamics, like the Gilbert damping, crucial for designing ultra-fast spintronic devices, remains largely unexplored. Despite recent studies by optical excitation and detection, achieving spin wave control with microwaves is highly desirable, as modern integrated information technologies predominantly are operated with these. The intrinsically small numbers of spins, however, poses a major challenge to this. Here, we present a hybrid approach to detect spin dynamics mediated by photon-magnon coupling between high-Q superconducting resonators and ultra-thin flakes of $Cr_2Ge_2Te_6$ (CGT) as thin as 11 nm. We test and benchmark our technique with 23 individual CGT flakes and extract an upper limit for the Gilbert damping parameter. These results are crucial in designing on-chip integrated circuits using vdW magnets and offer prospects for probing spin dynamics of monolayer vdW magnets.

van der Waals (vdW) materials[1–4] consist of individual atomic layers bonded by vdW forces and can host different types of collective excitations such as plasmons, phonons, and magnons. Strong coupling between these excitation modes and electromagnetic waves (i.e. photonic modes) creates confined light-matter hybrid modes, termed polaritons. Polaritons in vdW materials are an ideal model system to explore a variety of polaritonic states[5,6], e.g., surface plasmon polaritons in graphene[7,8] and exciton polaritons in a monolayer $MoS_2$ embedded inside a dielectric microcavity[9]. These states can be further modified by electrostatic gating[10], as well as by hetero-structuring with dissimilar vdW layers[1].

Numerous studies on magnon polaritons (MPs)[11,12] have been using macroscopic yttrium iron garnet (YIG) coupled to either three-dimensional cavities[13] or to on-chip resonators[14,15], with potential applications in ultra-fast information processing, non-reciprocity or microwave to optical transduction. By reducing the number of excitations, MPs find application in the quantum regime, e.g., magnon number counting via an electromagnetically coupled superconducting qubit[16,17] or as a building block for Bell state generation[18].

The rapidly developing research around polaritons and specifically MPs has so far, been little studied in magnetic vdW materials due to the relatively recent discoveries of long-range magnetic order in

[1]London Centre for Nanotechnology, University College London, 17-19 Gordon Street, London WCH1 0AH, UK. [2]Department of Physics, Faculty of Science, National University of Singapore, 2 Science Drive 3, Singapore 117542, Singapore. [3]National Physical Laboratory, Hampton Road, Teddington TW11 0LW, UK. [4]Institute for Condensed Matter Physics and Complex Systems, School of Physics and Astronomy, The University of Edinburgh, Edinburgh EH9 3FD, UK. [5]Higgs Centre for Theoretical Physics, The University of Edinburgh, Edinburgh EH9 3FD, UK. [6]Department of Electronic & Electrical Engineering, UCL, London WC1E 7JE, UK. [7]Centre for Advanced 2D Materials, National University of Singapore, 6 Science Drive 2, Singapore 117546, Singapore. [8]Department of Chemistry, Faculty of Science, National University of Singapore, 3 Science Drive 3, Singapore 117543, Singapore. [9]WPI Advanced Institute for Materials Research, Tohoku University, 2-1-1, Katahira, Sendai 980- 8577, Japan. ✉ e-mail: c.zollitsch@ucl.ac.uk

vdW systems at the few monolayer regime[19–21], in addition to its technically challenging realization. Stable MP states are formed by strongly coupling the magnetic field oscillation of a resonant photon to the collective magnetization oscillation in a magnetic material. This strong coupling is achieved when the collective coupling rate $g_{eff}$ is larger than the average of both system loss rates. In a simplified picture, $g_{eff}$ scales linearly with the strength of the oscillating magnetic field of a resonator and the square root number of spins[14]. For studies involving bulk magnetic materials and low-quality and large microwave resonators, strong coupling is achieved when $g_{eff}/2\pi$ is in the MHz range, which is accomplished with relative ease due to the abundance of spins in bulk magnetic materials. A reduction of the bulk dimensions down from mm to µm and nm scales, the typical lateral dimensions and thickness of vdW material monolayers, results in a decrease of the coupling strength by at least 6 orders of magnitude. Commonly used microwave resonators are not able to produce strong enough oscillating magnetic fields to compensate for such a reduction in absolute number of spins. Only by advanced resonator design and engineering the regime of strongly coupled MPs in monolayer vdW magnetic materials can be accomplished, granting access to spin dynamic physics at a true 2d monolayer limit and research on MPs in nanoscale devices where the whole range of on-chip tuning and engineering tools, such as electric fields or device design, are available.

Magnons or MPs have been observed in magnetic vdW materials, but it had been restricted to either the optical frequency range[22,23] or a large thickness limit[24,25], respectively. Here, we present our attempt of detecting spin dynamics in ultra-thin vdW magnetic materials and the creation of MPs by magnon-photon coupling in the microwave frequency range, using superconducting resonators optimized for increased magnon-photon coupling. By using microwave resonators with a small mode volume, we not only increased its oscillating magnetic field strength but also matched it more efficiently to the size of

nanoscale vdW flakes. Our work presents a fundamental cornerstone for a general blueprint for designing and developing magnon-photon hybrids for any type of ultra-thin or monolayer vdW magnetic material, enabling research on on-chip microwave applications for (quantum) information processing.

## Results

In this article, we report on the observation of spin dynamics and the creation of MPs at the onset of the high cooperativity regime with the vdW ferromagnet CGT of nm scale thickness, demonstrating a pathway towards stable magnon-photon polariton creation. We combine a precise transfer process of exfoliated CGT flakes and high-sensitivity superconducting resonators, to access and study the dynamical response of coupled photon-magnon states in a small-volume (nm-thick and µm-sized) CGT flake (illustrated in Fig. 1a). High-quality-factor superconducting lumped element resonators are chosen to be the counterpart due to their extremely small mode volume (≈6000 µm³) and consequently strong oscillating magnetic fields ($B_1 \approx 25$ nT, see SI for resonator quality-factors and $B_1$-field distributions), resulting in high spin sensitivities[26,27]. At cryogenic temperatures, we perform low-power microwave spectroscopy on multiple resonator-vdW-flake hybrids, covering a frequency range from 12 GHz to 18 GHz for a variety of thickness. Samples consist of up to 12 resonators on a single chip, all capacitively coupled to a common microwave transmission line for read-out (see SI for details). Multiple peaks of spin-wave resonances are observed for each CGT flake measured. The spin-wave modes are closely spaced in frequency and show a large overlap. We employ a semi-optimized fitting model to produce a good estimate for the collective coupling strength and magnetic linewidth. By taking the resonance value of the most prominent peak of each spectrum, we find that all measured points can be fitted very well by a single curve calculated by the Kittel formula with bulk CGT parameters. Furthermore, we extracted the linewidth for the thinnest CGT flake investigated, 11 nm or 15 monolayers (ML), the only device exhibiting well-separated spin-wave modes. This allowed a fully quantitative analysis and we determined an upper limit of the Gilbert damping parameter of 0.02. This value is comparable to the damping reported for 3d transition metal ferromagnets, suggesting that magnetic vdW flakes have the potential for the fabrication of functional spintronic devices.

We investigate the dynamics of nm-thick CGT flakes, using superconducting lumped element resonators made of NbN (see methods for fabrication details and SI and ref. 28 for more performance details). The advantages of a lumped element design are the spatial separation of the oscillating magnetic field $B_1$ and electric field $E_1$ and the concentration of $B_1$ within a narrow wire section of the resonators, as indicated in Fig. 1a. Additionally, the $B_1$ field distribution is homogeneous along the length of the narrow wire section (see finite element simulations in SI). This magnetic field concentration is our primary reason to use this type of resonator in order to reduce the photon mode volume as well as achieve a considerable mode overlap between the resonator photon mode and CGT magnon mode, and consequently, a large coupling strength. We, therefore, transfer CGT flakes onto these narrow sections (Fig. 1b). Details of CGT flake transfers are described in the methods section. Optical imaging and atomic force microscopy (AFM) measurements are used to characterize the size and thickness of the CGT flakes (see Fig. 1c). Measured thicknesses range from $153 \pm 23$ nm down to $11 \pm 1.8$ nm (15 ML), enabling a thickness-dependent study of CGT flakes and their coupling to the resonators.

We measured the microwave transmission $|S_{21}|^2$ as a function of frequency and externally applied magnetic field $B_0$ for each resonator at a temperature of 1.8 K, using a microwave power of ~−80 dBm at the resonator chip. Figure 2 (a) shows the resulting 2D plot of $|S_{21}|^2$ for a resonator loaded with a 17 nm ± 0.8 nm-thick CGT flake (see Fig. 1b, c

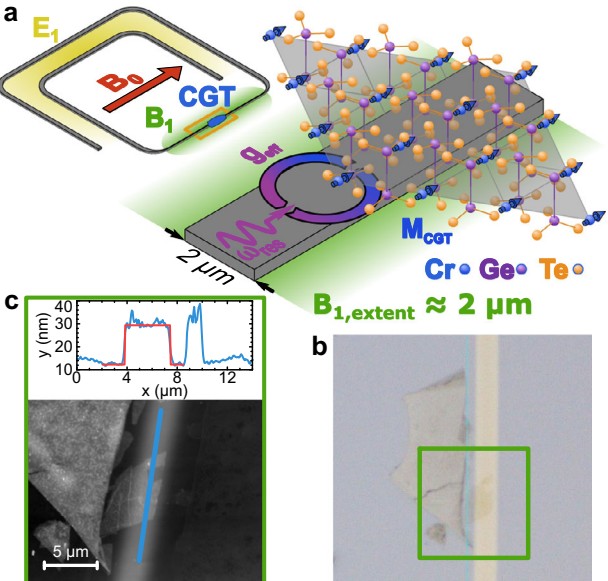

**Fig. 1 | Magnon-photon coupling between thin CGT and a superconducting resonator. a** Schematic of a resonator shows the design in detail, indicating the areas of high $E_1$-field (yellow) and $B_1$-field (green) intensities, as well as the orientation of the externally applied field $B_0$. Finally, a schematic zoom in of the section loaded with a CGT flake is shown. The collective coupling between a microwave photon and the magnetization of the CGT is illustrated, as well as the approximate extent of the microwave $B_1$-field. **b** Micrograph image of a CGT flake transferred onto the narrow section of a resonator. **c** AFM image of the CGT flake together with a height profile along the blue solid line in the AFM image. The red solid line is a fit to the flake thickness. The results of this resonator are presented in Fig. 2.

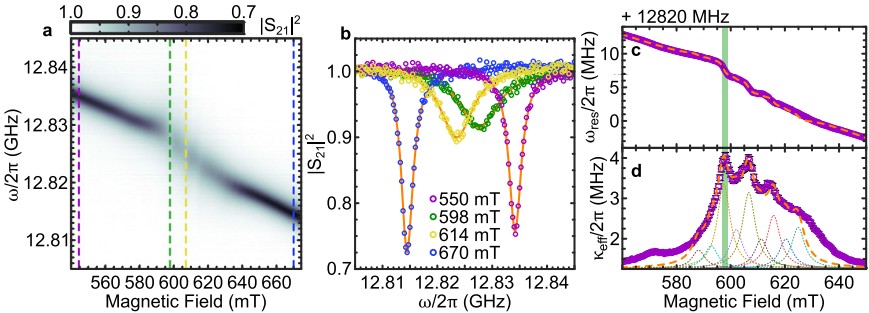

**Fig. 2 | Magnon-photon coupling observed in resonator microwave transmission. a** $|S_{21}|^2$ as a function static magnetic field $B_0$ and frequency, with the microwave transmission encoded in the color. The results are obtained from the resonator shown in Fig. 1b, c Featuring a loaded quality factor of $Q_L = 4600$. **b** $|S_{21}|^2$ as a function of frequency at fixed magnetic fields, indicated in **a** by dashed vertical lines. **c**, **d** Resonance frequency $\omega_{res}$ and effective loss rate $\kappa_{eff}$ as a function of magnetic field. Note the multiple resonance peaks, indicating multiple CGT FMRs. The dashed orange lines are results from the semi-optimized fit. **d** exemplary includes the individual peaks of which the orange dashed lines consists. The green bar in **c**, **d** highlights the main mode.

for the respective micrograph and AFM images). A resonator peak can be clearly observed for each magnetic field, with its resonance frequency $\omega_{res}$ decreasing with increasing magnetic field. The reduction of the frequency is a result of a slow degradation of the superconductivity by $B_0$, which in general exhibits a parabolic dependence[29]. For $580\,\text{mT} \leq B_0 \leq 630\,\text{mT}$ the resonator prominence is reduced, highlighted by $|S_{21}|^2$ as a function of frequency for four constant $B_0$ values in Fig. 2b. Within this field range, the mode resonance has been modified due to its hybridization with the magnetic modes of the CGT flake. To further quantify the interaction, we fit each $|S_{21}|^2$ profile by a Fano resonance lineshape (solid orange lines in Fig. 2b) to account for an asymmetric resonance peak due to additional microwave interference in the circuitry[30,31],

$$|S_{21}|^2 = S_0 + A\frac{\left(q\kappa_{eff}/2 + \omega - \omega_{res}\right)^2}{\left(\kappa_{eff}/2\right)^2 + \left(\omega - \omega_{res}\right)^2}. \quad (1)$$

Here, $S_0$ is the microwave transmission baseline, $A$ the peak amplitude, $q$ describes the asymmetry of the lineshape and $\kappa_{eff}$ represents the effective loss rate of the hybrid system (see SI for resonator parameters before and after CGT transfer for all resonators). Figure 2(c) shows $\omega_{res}$ of the hybrid system as a function of $B_0$. $\omega_{res}$ experiences a dispersive shift when the photon mode and the magnon mode hybridize, indicating an onset of a strong interaction between the two individual systems[14,17,32–34]. We observe multiple shifts in $\omega_{res}$, suggesting an interaction of several magnon modes with the resonator in our experiment.

Signatures of the resonator–CGT-flake coupling are also characterized by $\kappa_{eff}$ of the hybrid system (Fig. 2d). $\kappa_{eff}$ is enhanced from the value of the resonator loss rate $\kappa_0$ due to an additional loss introduced by the magnon system characterized by the loss rate $\gamma$[14,32,35]. Consistent with the $B_0$ dependence of $\omega_{res}$, $\kappa_{eff}$ shows a rich structure, having its main peak at 598 mT, together with less prominent peaks distributed around it. Based on a formalism for coupled-harmonic-oscillator systems in the high cooperativity regime[32–34], we use the following to analyse our experimental results with multiple peaks:

$$\omega_{res} = \omega_{res,0} + mB_0^2 + \sum_{k=-n}^{+n}\frac{g_{eff,k}^2\Delta_k}{\Delta_k^2 + \gamma^2}, \quad (2)$$

$$\kappa_{eff} = \kappa_0 + \sum_{k=-n}^{+n}\frac{g_{eff,k}^2\gamma}{\Delta_k^2 + \gamma^2}. \quad (3)$$

with the detuning factor for each resonance as $\Delta_k = \frac{g_{CGT}\mu_B}{h}\left(B_0 - B_{FMR,k}\right)$. Here, $\omega_{res,0}$ is the resonator resonance frequency at $B_0 = 0\,\text{T}$ and $m$ represents the curvature of the resonance

frequency decrease due to the applied magnetic field. $B_{FMR,k}$ is the CGT-FMR field, $g_{CGT}$ the g-factor of CGT and $g_{eff,k}$ gives the collective coupling strength between photon and magnon mode. The summation is over all resonance modes $k$ present on the low or high field (frequency) side of the main resonance mode, where $n$ gives the number of modes on one side. For simplicity, we assume a symmetric distribution of modes about the main mode. The large number of multiple modes and their strong overlap prevent a reliable application of a fully optimized fit to the data, due to the large number of free parameters required. In an effort to gain a good estimate of the model parameters we apply the model functions Eq. (2) and (3) in a two-step semi-optimized fashion (see SI for details). With this approach, we arrive at a model in good agreement with $\omega_{res}$ and $\kappa_{eff}$ (see orange dashed lines in Fig. 2c, d, exemplary showing the individual peaks of the orange dashed line in Fig. 2d and the SI for additional results and data). We can reproduce the data using $\gamma/2\pi = 94.03 \pm 5.95\,\text{MHz}$ and a collective coupling strength of the main mode of $13.25 \pm 1\,\text{MHz}$. Together with $\kappa_0/2\pi = 1.4 \pm 0.02\,\text{MHz}$ the system resides at the onset of the high cooperativity regime, classified by the cooperativity $C = g_{eff}^2/\kappa_0\gamma = 1.3 > 1$[13,32]. In this regime, MPs are created and coherently exchange excitations between magnons and resonator photons on a rate given by $g_{eff}$. The created MPs are, however, short-lived and the excitations predominately dissipate in the magnonic system, as $g_{eff} \ll \gamma$.

Our analysis suggests that the separation of the different FMR modes is of the same order of magnitude as the loss rate (see SI for additional data). We consider that these are from standing spin-wave resonances, commonly observed for thin magnetic films[36] and with one reported observation in bulk of the vdW material $CrI_3$[37]. In thin-film magnets under a static magnetic field applied in-plane, the magnetic-dipole interaction generates two prominent spin-wave branches for an in-plane momentum, the backward volume spin-wave (BVMSW) and magnetostatic surface spin-wave (MSSW) modes[38,39]. These spin-wave modes have different dispersion relations, having higher (MSSWs) and lower (BVMSWs) resonance frequencies with respect to that of the uniform FMR mode. We calculate the distance of these standing spin-wave modes based on magnetic parameters of bulk CGT as well as the lateral dimensions of the flakes (see SI for more details). We can find spin waves having a frequency separation within 100 MHz and 200 MHz (3.3 mT to 6.6 mT in magnetic field units), which are consistent with our experimental observation in terms of its mode separation. However, the irregular shape of the CGT flakes renders exact calculations of spin-wave mode frequencies very challenging. We also considered the possibility that each layer of CGT might have different magnetic parameters (e.g., chemical inhomogeneity), and thus producing different individual resonance modes. Our numerical simulations based on atomistic spin dynamics[40,41] rule out this

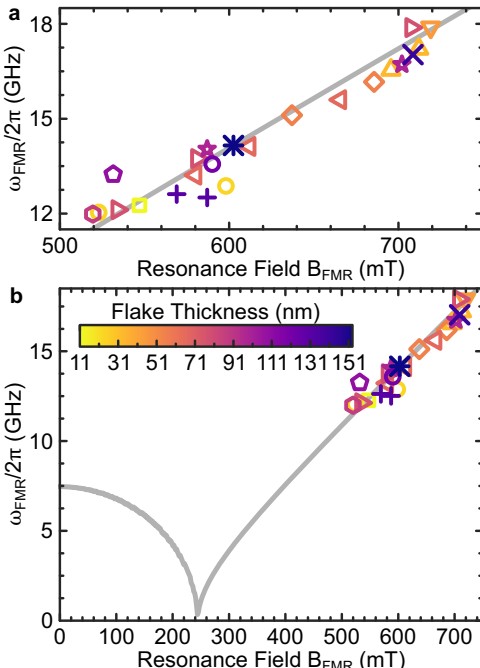

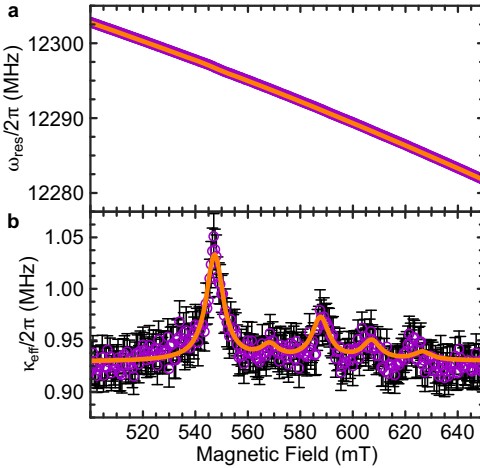

**Fig. 3 | Summary of CGT-FMR conditions. a** Extracted CGT resonance fields and frequencies from the set of resonators loaded with CGT flakes of different thickness. Resonance values are taken from the most prominent peaks in $\kappa_{\text{eff}}$. The solid curve is calculated using the Kittel formalism presented in ref. 42, using same parameters, with $g_{\text{CGT}} = 2.18$, $\mu_0 M_s = 211.4$ mT, and $K_u = 3.84 \times 10^4$ J/m³. **b** Wider magnetic field range of **a** where the CGT flake thickness for the different symbols is indicated by the color gradient given in **a**.

**Fig. 4 | Magnon-photon coupling for the thinnest CGT flake. a** Resonance frequency $\omega_{\text{res}}$ and **b** effective loss rate $\kappa_{\text{eff}}$ as a function of the magnetic field of a resonator loaded with the thinnest CGT, consisting of 15 ML. The resonator's loaded quality factor is 6938. The solid orange lines are the results a fit to Eq. (2) and (3), respectively. The error bars in **b** represent the standard deviation from the Fano resonance lineshape fit to the resonator transmission.

possibility, as resonance modes from individual layers average to a single mode as soon as a fraction of 10% of inter-layer exchange coupling is introduced (see SI for more details). Therefore, we speculate that the multiple-mode nature we observe in our experiments is likely originating from intrinsic properties of the CGT flakes.

Figure 3 shows the extracted $\omega_{\text{FRM}}$ as a function of $B_{\text{FMR}}$ for each resonator–CGT-flake hybrid. The experimental values are in excellent

agreement with a curve calculated by the Kittel equation with magnetic parameters for bulk CGT[42], from which the data exhibits a standard deviation of <5 %. This agreement, achieved by independent characterizations of 23 CGT flakes measured by superconducting resonators, is experimental evidence that the magnetic parameters that determine the dispersion of $\omega_{\text{FRM}}(B_{\text{FMR}})$, i.e., the CGT g-factor $g_{\text{CGT}}$, saturation magnetization $M_s$ and uniaxial anisotropy $K_u$, exhibit little thickness dependence in exfoliated CGT flakes, and are not disturbed by the transfer onto the resonator structure. We note, that this demonstrates that vdW magnetic materials are particularly attractive for device applications, as they are less prone for contamination from exfoliation.

Finally, we present our analysis of $\kappa_{\text{eff}}$ for a resonator with a $11 \pm 1.8$ nm CGT flake in Fig. 4. With the thickness of a single layer of CGT being 0.7 nm[19], this flake consists of 15 monolayers and is the thinnest in our series. Figure 4a, b show $\omega_{\text{res}}$ and $\kappa_{\text{eff}}$ as a function of $B_0$, respectively. While the response of the CGT flake shows a prominent signature in $\kappa_{\text{eff}}$, the CGT-FMR is considerably more subtle in $\omega_{\text{res}}$. This highlights the excellent sensitivity of the high-Q superconducting resonators in our study. $\kappa_{\text{eff}}$ features five well-separated peaks with the main peak at $B_0 = 547$ mT, which enables us to perform a single-peak fully optimized analysis for each, in contrast to our multi-step analysis for the remainder of the devices. We assume the additional peaks are BVMSW modes, as discussed in the previous section. However, the splitting is about four times larger than compared to all other investigated devices, which would result in a significantly shorter wavelength. Thickness steps can lead to a wavelength down-conversion[43], however, due to the irregular shape and $B_1$ inhomogeneities it is difficult to exactly calculate the spin-wave frequencies (see SI for further details). From the main peak profile, we extract $g_{\text{eff}}/2\pi = 3.61 \pm 0.09$ MHz, $\gamma/2\pi = 126.26 \pm 8.5$ MHz and $\kappa_0/2\pi = 0.92 \pm 0.05$ MHz. We compare the experimental value of $g_{\text{eff}}$ with a numerically calculated $g_{\text{eff,simu}}$, using the dimensions of the CGT flake determined by AFM measurements (see SI for details). The calculation yields $g_{\text{eff,simu}}/2\pi = 8.94$ MHz, lying within the same order of magnitude. The overestimation is likely due to in-perfect experimental conditions, like non-optimal placement of the flake, uncertainties in the thickness and dimension determination as well as excluding the additional modes in the calculation (see SI). With $\gamma \gg g_{\text{eff}}$ and $C = 0.11$, the hybrid system is in the weak coupling regime[13], but due to the highly sensitive resonator with its small $\kappa_0$ the response from the magnon system can still be detected. With the extracted $\gamma/2\pi$ we can give an upper limit of the Gilbert damping in CGT, by calculating $\alpha_{\text{upper}} = \gamma/\omega_{\text{FMR}}$. We find $\alpha_{\text{upper}}$ as $0.021 \pm 0.002$, which is comparable to other transition metal magnetic materials[44], and is in very good agreement with a previously reported effective Gilbert damping parameter determined by laser-induced magnetization dynamics[45]. Here, we emphasize that the actual Gilbert damping value is lower due to a finite, extrinsic inhomogeneous broadening contribution.

We further use these results to benchmark the sensitivity of our measurement techniques. The detection limit is given by comparing the main peak height characterized by $g_{\text{eff}}^2/\gamma$ and the median noise amplitude which is 18 kHz in Fig. 4b where $g_{\text{eff}}^2/2\pi\gamma = 103$ kHz. By assuming the same lateral dimensions and scale the thickness down to a single monolayer, while keeping $\gamma$ constant, we calculate the expected signal reduction numerically by $g_{\text{eff,simu,1ML}}/g_{\text{eff,simu,15ML}}$ to 0.26. We obtain $(0.26 g_{\text{eff}})^2/2\pi\gamma = 7$ kHz for the monolayer limit. Although this suggests the noise amplitude is greater than the expected peak amplitude, we can overturn this condition by improving the coupling strength by optimizing the resonator design, enhancing the exfoliation and flake transfer as well as by reducing the noise level by averaging a number of multiple scans. Superconducting resonators with mode volumes of ~10 μm³ have been realized[46], a reduction of 2 orders of magnitude compared to our current design. This would translate to an order of magnitude improvement in $g_{\text{eff}}$. Furthermore, this flake covers ~4% of the resonator. By assuming

maximized coverage a five times enhancement of $g_{eff}$ can be achieved. Both approaches would make the detection of monolayer flakes possible.

In summary, we provide the first demonstration of photon-magnon coupling between a superconducting resonator and nm-thick vdW flakes of CGT, using a total of 23 devices with different CGT flakes of thickness from 153 nm down to 11 nm. By employing a coupled-harmonic-oscillator model, we extract the coupling strength, magnetic resonance field, and relaxation rates for both photon and magnon modes in our devices. From our semi-broadband experiments, we find that the magnetic properties of exfoliated CGT flakes are robust against the transfer process, with a standard deviation of less than 5 % to expected resonance values from bulk parameters. Notably, this suggests that vdW magnetic materials can be pre-screened at bulk to identify the most promising material for few-layer device fabrication. The upper limit of the Gilbert damping in the 15 ML thick CGT flake is determined to be 0.021, which is comparable to commonly used ferromagnetic thin-films such as NiFe and CoFeB and thus making CGT attractive for similar device applications. We highlight that the damping parameter is key in precessional magnetization switching[47,48], auto-oscillations by dc currents[49,50], and comprehensive spin-orbit transport in vdW magnetic systems[51]. The presented techniques are readily transferable to other vdW magnetic systems to study spin dynamics in atomically-thin crystalline materials. While creating stable MPs is still an open challenge due to the large loss rate $\gamma$ of the CGT magnon system, this work offers an important approach towards its achievement. There are still potential improvements to the measurement sensitivity such as resonator mode volume reduction by introducing nm scale constrictions[52,53] and use of exfoliation/transfer techniques to produce larger flakes to enhance the mode overlap (hence coupling strength)[54,55]. With concerted efforts, the formation of MPs in few layers vdW materials will become feasible.

## Methods

### Superconducting resonators

The resonators were fabricated by direct laser writing and a metal lift-off process. The individual 5 mm × 5 mm chips are scribed from an intrinsic, high resistivity ($\rho > 5000\ \Omega$cm) n-type silicon wafer of 250 μm thickness. For a well-defined lift-off, we use a double photoresist layer of LOR and SR1805. The resonator structures are transferred into the resist by a Heidelberg Direct Writer system. After development, ~50 nm NbN are deposited by magnetron sputtering in a SVS6000 chamber, at a base pressure of $7 \times 10^{-7}$ mbar, using a sputter power of 200 W in an 50:50 Ar/N atmosphere held at $5 \times 10^{-3}$ mbar, with both gas flows set to 50 SCCM[28]. Finally, the lift-off is done in a 1165 solvent to release the resonator structures.

### CGT crystal growth

CGT crystals used in this study were grown via chemical vapor transport. To this end, high-purity elemental precursors of Cr (chips, ≥ 99.995%), Ge (powder, ≥ 99.999%), and Te (shots, 99.999%) were mixed in the molar weight ratio Cr:Ge:Te = 10:13.5:76.5, loaded into a thick-wall quartz ampule and sealed under the vacuum of ~$10^{-5}$ mbar. Then, the ampule was loaded into a two-zone furnace, heated up and kept at 950° for 1 week to homogenize the precursors. To ensure high-quality growth, the ampule was slowly cooled (0.4°/h) maintaining a small temperature gradient between the opposite ends of the ampule. Once the ampule reached 500°, the furnace was turned off allowing the ampule to cool down to room temperature naturally. The large (~1 cm) single-crystalline flakes were extracted from the excess tellurium and stored in an inert environment.

### CGT flake transfer

Devices for this study were made via the transfer of single-crystalline thin flakes on top of the superconducting resonators. The flakes were first exfoliated from bulk crystals on the clean surface of a home-cured PDMS (polydimethylsiloxane, Sylgard 184) substrate. The thickness of the CGT flakes on PDMS was estimated through the contrast variation with transmission optical microscopy. Then, the selected flake was transferred to a resonator. The transfer was performed in air at room temperature. To minimize the air exposure, the entire process of exfoliation, inspection, and transfer was reduced to 10–15 min per resonator. For the flakes thicker than 50 nm, the strong optical absorption of CGT prevented the accurate thickness estimation with optical contrast. For those flakes, the thickness was estimated via a quick AFM scan performed on the PDMS substrate before the transfer step. Ready devices were stored in inert conditions.

## Data availability

The data that support the findings of this study are available within the paper, Supplemental Material and from the corresponding authors upon reasonable request.

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

## Acknowledgements

This study is supported by EPSRC on EP/T006749/1 and EP/V035630/1. G.E. acknowledges support from the Ministry of Education (MOE), Singapore, under AcRF Tier 3 (MOE2018-T3-1-005) and the Singapore National Research Foundation for funding the research under the medium-sized center program. E.J.G.S. acknowledges computational resources through CIRRUS Tier-2 HPC Service (ec131 Cirrus Project) at EPCC funded by the University of Edinburgh and EPSRC (EP/P020267/1); ARCHER UK National Supercomputing Service (http://www.archer.ac. uk) via Project d429. EJGS acknowledges the Spanish Ministry of Science's grant program "Europa-Excelencia" under grant number EUR2020-112238, the EPSRC Early Career Fellowship (EP/T021578/1), and the University of Edinburgh for funding support. D.S. acknowledges EPSRC funding through the Center for Doctoral Training in Advanced Characterization of Materials (EP/L015277/1) and European Union's Horizon 2020 Research and Innovation program under grant agreement GrapheneCore3, number 881603 and the Department for Business, Energy and Industrial Strategy through the NPL Quantum Program.

## Author contributions

C.W.Z., S.K., and H.K. conceived the experimental project. Resonator design and optimization were done by J.O'S., O.W.K, C.W.Z., and supervised by J.J.L.M. Resonator fabrication and characterization were done by C.W.Z. CGT crystals were grown by I.A.V. and exfoliated and transferred by I.A.V. and V.T.T.N. and supervised by G.E. D.S. measured AFM on the CGT flakes on the resonators. C.W.Z. performed the experiments and the data analysis with input from S.K. and H.K. Atomistic spin dynamics simulations were carried out by M.S. supervised by E.J.G.S. C.W.Z., M.S., I.A.V., and H.K. wrote the manuscript with input from all authors.

## Competing interests

The authors declare no conflict of interest.
