## [Peer Review File · Nature Communications]

Reviewers' Comments:

Reviewer #1:

Remarks to the Author:

In the current version, the authors attempted to improve the manuscript with some minor revisions. But, these minor revisions do not settle my concern. Most importantly, some new physics or application is still missing as a consequence of using the superconducting resonators and the CrGeTe thin-film flakes, while the authors claimed that "we address directly the challenge of detecting spin dynamics in ultra-thin vdW magnetic materials and the creation of MPs by magnon-photon coupling in the microwave frequency range,..." Therefore, I would suggest the authors to further highlight the significance with some new experimental results before I could recommend it to be published in Nature communications.

Reviewer #2:

Remarks to the Author:

The authors addressed my previous questions. The presence of multiple modes is, as the authors stated, still an open question. The additional data analysis didn't provide strong supporting evidence for assigning them to BVMSW modes. But the report itself can be helpful for future groups to look into this problem. This paper can be published in Nature Communication.

Response Letter for NCOMMS-22-52698-T

We thank all reviewers for their reports and valuable comments and suggestions on our work. We respond to the points raised by the reviewers below.

Reviewer #1 (Remarks to the authors):

In the current version, the authors attempted to improve the manuscript with some minor revisions. But, these minor revisions do not settle my concern. Most importantly, some new physics or application is still missing as a consequence of using the superconducting resonators and the CrGeTe thin-film flakes, while the authors claimed that "we address directly the challenge of detecting spin dynamics in ultra-thin vdW magnetic materials and the creation of MPs by magnon-photon coupling in the microwave frequency range,..." Therefore, I would suggest the authors to further highlight the significance with some new experimental results before I could recommend it to be published in Nature communications.

Response: We thank the Reviewer for his/her comments on our manuscript. We would like to highlight that the significance of our study is to present a first strong step to show the methodology of magnon polaritons, using a van der Waals / superconducting-resonator hybrid system. We present a quantitative analysis on this methodology and test its applications over a wide range of devices with resonators having different resonance frequencies and different thicknesses of CGT flakes. This thorough test shows the validity of such an approach and we present it as a promising platform for studying spin dynamics and magnon polaritons in thin van der Waals magnets. Finally, we provide suggestions for further improvements to realize monolayer spin dynamics detection and/or the clear formation of magnon polaritons.

To better emphasize that our study is providing a fundamental methodology to a separate new research field of hybrid van der Waals magnon-photon devices, we rewrite the quoted sentence in line 77:

"Here, we address directly the challenge of detecting spin dynamics in ultra-thin vdW magnetic materials and the creation of MPs by magnon-photon coupling in the microwave frequency range, using superconducting resonators optimized for increased magnon-photon coupling."

to

*"Here, we **present our attempt** of detecting spin dynamics in ultra-thin vdW magnetic materials and the creation of MPs by magnon-photon coupling in the microwave frequency range, using superconducting resonators optimized for increased magnon-photon coupling."*

Reviewer #2 (Remarks to the authors):

The authors addressed my previous questions. The presence of multiple modes is, as the authors stated, still an open question. The additional data analysis didn't provide strong supporting evidence for assigning them to BVMSW modes. But the report itself can be helpful for future groups to look into this problem. This paper can be published in Nature Communication.

Response: We thank the Reviewer for reviewing our manuscript, and suggesting its publication. We further thank the reviewer for stating that our work will be helpful to other groups, and we hope it will spark an increased focus on this research area.